# Estimation of Pediatric Dosage of Antimalarial Drugs, Using Pharmacokinetic and Physiological Approach

**DOI:** 10.3390/pharmaceutics15041076

**Published:** 2023-03-27

**Authors:** Ellen K. G. Mhango, Bergthora S. Snorradottir, Baxter H. K. Kachingwe, Kondwani G. H. Katundu, Sveinbjorn Gizurarson

**Affiliations:** 1Faculty of Pharmaceutical Sciences, School of Health Sciences, University of Iceland, 107 Reykjavik, Icelandbss@hi.is (B.S.S.); 2Department of Pharmacy, School of Life Sciences and Allied Health Professions, Kamuzu University of Health Sciences, P/Bag 360, Blantyre 3, Malawi; 3Biomedical Sciences Department, School of Life Sciences and Allied Health Professions, Kamuzu University of Health Sciences, P/Bag 360, Blantyre 3, Malawi

**Keywords:** dosage, pharmacokinetics, malaria, antimalarial

## Abstract

Most of the individuals who die of malaria in sub–Saharan Africa are children. It is, therefore, important for this age group to have access to the right treatment and correct dose. Artemether—lumefantrine is one of the fixed dose combination therapies that was approved by the World Health Organization to treat malaria. However, the current recommended dose has been reported to cause underexposure or overexposure in some children. The aim of this article was, therefore, to estimate the doses that can mimic adult exposure. The availability of more and reliable pharmacokinetic data is essential to accurately estimate appropriate dosage regimens. The doses in this study were estimated using the physiological information from children and some pharmacokinetic data from adults due to the lack of pediatric pharmacokinetic data in the literature. Depending on the approach that was used to calculate the dose, the results showed that some children were underexposed, and others were overexposed. This can lead to treatment failure, toxicity, and even death. Therefore, when designing a dosage regimen, it is important to know and include the distinctions in physiology at various phases of development that influence the pharmacokinetics of various drugs in order to estimate the dose in young children. The physiology at each time point during the growth of a child may influence how the drug is absorbed, gets distributed, metabolized, and eliminated. From the results, there is a very clear need to conduct a clinical study to further verify if the suggested (i.e., 0.34 mg/kg for artemether and 6 mg/kg for lumefantrine) doses could be clinically efficacious.

## 1. Introduction

Malaria is a parasitic disease that is disseminated by the bite of an infected female mosquito [1,2]. In 2019, approximately 409,000 people perished because of malaria. Approximately 67% of these were children and 94% of the cases and mortalities happened in Africa. In 2018, out of the deaths that occurred due to malaria (405,000), 94% occurred in sub–Saharan Africa [2,3,4]. Fever, headache, vomiting, pain in joints, and chills are the symptoms of malaria [5]. Cerebral malaria, low glucose levels, and anemia are frequent characteristics of severe malaria more noticeable in children than in grown-ups [3]. The vulnerability of children to respiratory infections, diarrhea, and other sicknesses rises when they develop recurrent malaria [3,6].

Unfortunately, most children do not survive these life-threatening afflictions. According to the World Health Organization (WHO) report in 2021, children’s deaths increased by 69,000 in sub-Saharan Africa alone during the COVID-19 pandemic due to disruptions in the provision of malaria prevention, diagnosis, and treatment [7]. These life-threatening situations require immediate treatment. Therefore, in these circumstances, it is important to have access to treatment that can be used by everyone, with the right dose for infants and small children.

### 1.1. Pharmacokinetics

Traditionally, dosages for children have been based on extrapolation of adult dosing using the standardized body weight of grown-ups (70 kg) or body surface area (1.73 m^2^). Several methods have been suggested based on using an adult dose and correcting it using different factors based on the child’s age, weight, and/or its body surface area. Examples of these equations are Young’s equation, Clark’s equation, Crawford’s equation, etc. [8,9]. Guidelines published by the WHO on how to dose antimalarial drugs in children are also based on body weight [10]. However, children under the age of 5 years are not small adults, and their physiology, liver enzyme maturation, and serum proteins have not fully developed and may, therefore, significantly influence the pharmacokinetics of drugs and how they behave in the body.

Therefore, the use of these approaches may not provide the optimal treatment in small children and may result in either underdosing or overdosing. This is not only the case for antimalarial drugs but also for many other pediatric treatments for which the proposed dosage regimens are centered on findings from grown-up patients. The impact of underdosing on antimalarial management failure has not been recognized much [11,12]. Young children and pregnant women suffer the greatest burden of malaria. Therefore, the administered dose must be able to deliver adequate concentrations to kill the parasite, cure the child, and secure the safety of the drugs used [11].

Allometric scaling is an experimental methodology often used for scaling between animal species, where the conversion of dose is based on the surface area of the body or on the weight associated with the metabolic rate [12]. The usual purposes of interspecies scaling involve predicting the dose in humans as a result of extrapolating from preclinical experiments [10].

During development from fetus to adulthood, the human body undergoes physiological changes that affects the body’s absorption, distribution, metabolism, and excretion (ADME) of both endogenous compounds as well as xenobiotics [13,14,15]. In oral administration of drugs, the developmental changes that influence absorption include gastric pH, gastric emptying time and motility, intestinal wall enzymes and transporters, variations in blood perfusion, the gastrointestinal microbiome, and environmental factors such as food (e.g., milk). Bile acids, biliary function, and pancreatic enzymes are essential in the absorption of lipophilic molecules, drugs, and drug esters from the GIT through solubilization and cleavage of prodrugs [15,16]. The amount and composition of bile acids are low in neonates and continue to increase during development to reach adult levels [16,17,18].

In relation to comorbidities, the drugs used to treat comorbidities may alter the pharmacokinetics of AR-LF or cause adverse drug reactions; for example, ketoconazole (an antifungal drug) has been reported to increase the plasma concentration of AR-LF through inhibition of the enzyme (CYP3A44) responsible for metabolizing these drugs [19]. In individuals coinfected with malaria and HIV, lopinavir/ritonavir raises AR exposure but decreases LF exposure [20]. Efavirenz also decreases AR-LF exposure, while nevirapine reduces exposure of AR but LF remains unaffected. However, AR-LF decreases exposure of nevirapine [21]. Coadministration of AR-LF with other antimalarial drugs, such as quinine, decreases the exposure of AR but has no effect on LF [22]. Meanwhile, the antimalarial mefloquine does not alter the PK of AR, but it decreases the plasma concentration of LF [23].

Rifampicin, a drug used in tuberculosis treatment, decreases plasma concentrations of AR-LF [24]. Foods such as grape juice and grapefruit raise the concentration of AR-LF [25]. These examples of drug—drug interactions or food—drug interactions may elevate the risk of toxicity or therapy failure and development of resistance.

### 1.2. Plasma Proteins

Many drugs bind to plasma proteins to a certain degree. Drug distribution and elimination are altered by plasma protein binding, which is influenced by different factors such as the concentration of proteins in the body, affinities between drugs and proteins, hepatic or renal diseases, and the physicochemical properties of the drug. Albumin, α_1_-acid glycoprotein (AAG), and lipoproteins are the main proteins to which drugs bind [26]. Plasma albumin is the key protein (35–50 g/L), as shown in Table 1, and is responsible for binding many acidic (anionic) and basic drugs. Since the synthesis of albumin takes place in the liver, its concentration may be lowered when the liver is suffering due to disease, helminthics, or conditions such as malnutrition [26,27].

Although AAG is present in much low concentrations (1–3 g/L, see Table 1), it is responsible for binding to many basic (cationic) and neutral drugs. Like albumin, the synthesis of AAG takes place in the liver. Contrary to albumin, medical conditions such as malaria causes AAG concentrations to elevate [26,28]. Likewise, the synthesis of lipoproteins, which comprise high density lipoproteins (HDL), low density lipoproteins (LDL), and very low-density lipoproteins (VLDL), takes place in the liver and intestinal mucosa (Table 1). They are known to bind to very lipophilic basic and neutral drugs [26]. Values used in Table 1 for lipoproteins were extracted from Bogulasa heart study [29], values for malnourished children were extracted from Barroso et al. [30] and values for serum lipids were extracted from Carvajal et al. [31].

Children are known to have lower concentrations of plasma proteins, as shown in Table 1. This may affect the tissue distribution of drugs, the free (unbound) drug concentration, as well as influence the clearance. Some drugs are known to bind to other plasma components, such as lipoproteins and red blood cells, and their uptake from plasma is usually concentration dependent. Since the malaria parasite hides inside red blood cells, this could be beneficial for the efficacy of the drug. However, having a high free drug concentration may have its drawbacks, such as altered volume of distribution, clearance, and half-life, since free drug is cleared more easily than drug bound to plasma proteins and red blood cells.

The WHO recommended the use of AR and LF as one of the artemisinin-based combination therapies (ACTs) to treat uncomplicated *Plasmodium falciparum* malaria [32]. Therefore, AR and LF were chosen as the model drugs in this work. Artemether works by forming free radicles that delay the growth of proteins during development of the parasites, while LF works by detoxifying hemoglobin in the parasite [33,34]. Lumefantrine is a very highly protein bound drug (99.7%), leaving only 0.3% free concentration of active drug [35]. It binds primarily to HDL (77%), LDL (7.3%), and VLDL (6.6%). The fraction of LF that binds to albumin and AAG is negligible. The fraction that binds to red blood cells is only 8% [36].

The magnitude of protein binding may not be affected by the condition the disease (malaria) causes. LF is most likely metabolized to a low degree in vivo, indicating that it can be categorized as a capacity-limited (poorly extracted) binding-sensitive drug (i.e., a drug that is highly bound with a low hepatic extraction ratio) [37]. AR is also highly bound to proteins (95.4%) and has, therefore, a much higher free concentration (4.6%) [35]. AR binds mainly to AAG (33%), albumin (17%), HDL (12%), LDL (9.3%), and VLDL (12%). The fraction that binds to red blood cells is approximately 11% [36].

### 1.3. Metabolism

Although metabolism can take place in various areas of the body, such as the kidney, lungs, gastrointestinal tract, and placenta, the liver is the main site of metabolism, and the expression of enzymes is different among infants, children, and grown-ups. Because of this, metabolism varies with age. Therefore, it is challenging to predict safe and effective pediatric drug dose using data from adults only [8,38]. Another factor that needs to be taken into consideration is that the size of the liver in infants and small children is relatively larger than that in adults. This may be reflected by variations in the activity of metabolizing enzymes with respect to liver size. Having a relatively larger organ will require relatively more blood flow, which will affect the disposition and can lead to either underexposure and treatment failure or overexposure with accompanying toxicity [39].

### 1.4. Nutritional Status

The relationship between undernutrition and malaria remains complex, with other links deemed controversial and others in the course of being delineated [40,41]. Overall, however, undernutrition in children under five years increases the risk of malaria, other infections, and recurrent malaria parasitemia [42,43,44,45]. Moreover, chronic undernutrition is a determinant of severe malaria complications [40,46] and being underweight contributes to approximately double the risk of malaria re-infection following successful treatment [47,48]. The effect of malnutrition on malaria and its treatment is attributed to the pathophysiology driven by malnutrition.

Severe acute malnutrition (SAM) results in pathophysiological alterations in tissue, organs, and the body in general [49]. Various physiologic adaptations in SAM include growth restriction, loss of fat, muscle, and visceral mass, reduced basal metabolic rate, and reduced total energy expenditure [50,51]. Organ systems are also variably impaired in SAM, including the immune system, in which cellular immunity is impaired and the body is rendered susceptible to infections [50,51]. Chronic undernutrition may increase the risk of severe malaria and re-infection due to the altered immune response of the host [40]. Undernutrition may also obscure the overt signs and symptoms of clinical malaria usually observed in children with a normal nutrition status [40,41].

Another significant pathophysiological alteration in malnourished children is intestinal villous atrophy in the jejunal mucosa, with resultant malabsorption of nutrients and drug substances [49]. Figure 1 shows the anatomical alterations in the intestinal mucosa as a result of severe malnutrition and the interaction with bile stimulation differentiating the responses between the two anatomically different mucosae.

The purpose of this article was to estimate the dosage for the two antimalarial drugs, artemether and lumefantrine, in young children under 5 years of age using a physiological and pharmacokinetic approach.

## 2. Methods

### 2.1. Protein Binding

There are many physiological and anatomical factors that undergo transformation when newborns become infants and then grow to become children. Some of these factors are not fully developed until children are around 18 years and many of them will influence the pharmacokinetics of drugs. Predicting the free drug fraction in neonates and infants has been studied by McNamara and Alcorn [52]. Knowing the value of the unbound drug fraction in plasma (*f_u_*) can be used to predict the free fraction in infants and children [1], as shown in Equation (1). The unbound free fraction of drugs in children (*fu_child_*) can be estimated from the free fraction in adults (*fu_adult_*) and the protein concentration [*P*] ratios between adults and children, using Equation (1) [1].
(1)$${fu}_{child}=\frac{1}{1+\frac{{\left[P\right]}_{child}}{{\left[P\right]}_{adult}}\times \frac{\left(1-{fu}_{adult}\right)}{{fu}_{adult}}}$$

### 2.2. Metabolism and CYP Enzyme Maturation

The maturation of enzymes takes time and has been expressed as the ratio of adult abundance based on age. The age-related maturation of enzymes that are important for artemether (CYP2B6 and CYP3A4) and lumefantrine (CYP3A4) is expressed as [53]:(2)$$CYP2B6=\frac{1.07\times Age}{1.13+Age}$$

For the enzyme CYP3A4, the maturation rate changes around 2.3 years, so for children below the age of 2.3 years, the following maturation model should be used [54]:(3)$$CYP3A4=0.11+\frac{{0.95\times Age}^{1.91}}{0.64^{1.91}+{Age}^{1.91}}$$

Then, for children 2.3 years and older, the maturation rate follows [54]:(4)$$CYP3A4=1.1-0.123\times e^{\left(-0.05\left(Age-2.2\right)\right)}$$

Using the enzyme maturation model incorporating the enzymes involved for each compound, such as artemether, the overall degree of enzyme maturation (MFA) in children below the age of 25 years is calculated by adding the outcomes into the following equation [8], such as for artemether [54]:(5)$$MFA=\left(0.111\times CYP2B6\right)+\left(0.889\times CYP3A4\right)$$

This information may then be used to calculate the estimated clearance in children using the following equation. Note that this equation does not account for differences in the volume of distribution in children compared to that in adults [54]:(6)$${Cl}_{children}={Cl}_{adults}{\left(\frac{Liver\;weight(child)}{Liver\;weight(adult)}\right)}^{0.75}\times MFA$$

### 2.3. Dosage Calculation

The equations that can be used to extrapolate the adult dosage to an equivalent dose in children are shown below.

#### 2.3.1. Young’s Rule

Young described the relationship between adult and child doses using the following equation, in which he only focused on the child’s age [55]:(7)$$D_{child}=average\;adult\;dose\left(\frac{age\;of\;child\left(y\right)}{age\;of\;child\left(y\right)+12}\right)$$
where *D_child_* is the dosage in a child.

#### 2.3.2. Clark’s Rule

Clark described the relationship between adult and child doses using the following equation, in which he only focused on the weight of the child and compared that with the equivalent dose in a 68 kg adult [55]:(8)$$D_{child}=adult\;dose\left(\frac{weight\;of\;child\left(kg\right)}{68\;kg}\right)$$

#### 2.3.3. Area Rule

Few researchers have focused on the body surface area (BSA) as the method to calculate the dose [55]:(9)$$D_{child}=adult\;dose\left(\frac{BSA\;of\;child\left(m^{2}\right)}{1.73\;m^{2*}}\right)$$

Here, the BSA can be calculated using the Mosteller formula, where W is the weight in kg and H is the height in cm [56]:(10)$$BSA=\sqrt{H\times \frac{W}{3600}}$$

#### 2.3.4. Other Method Similar to Allometric Method

Another method that has been used by some researchers is similar to allometric scaling, in which the adult weight is proportional to the child weight to the 0.75 power [57], as shown below:(11)$$D_{child}=adult\;dose\times 1.5{\left(\frac{weight\;of\;child\;(kg)}{70\;kg}\right)}^{\frac{3}{4}}$$

A variation of this method was introduced by Mahmood [58], who suggested the power varies depending on the child’s age. This author used the following formula to estimate the clearance in children:(12)$${Cl}_{pediatric}={Cl}_{adult}\times {\left(\frac{{weight}_{pediatric}}{{weight}_{adult}}\right)}^{exponent}$$
where the exponents are 0.75 for children older than 5 years, 0.9 for children 2–5 years, 1.0 for infants 3 months to 2 years, and 1.2 for infants 1–3 months.

### 2.4. Pharmacokinetic Method

The pharmacokinetic parameters for artemether and lumefantrine listed in Table 2, were obtained from WHO and values for children and malnourished children were extracted from different articles [19,59]. When the pharmacokinetic method is used, the dose is calculated based on the desired plasma concentration. Here, the desired concentration may be based on the plasma concentration reached when an adult receives a dose. At any time, point, the plasma concentration (*C_p_*) in an adult may be calculated using adult dose and pharmacokinetic parameters for these antimalarial drugs in adults, using the following equation:(13)$$C_{p}=\frac{FDk_{a}}{V_{d}\left(k_{a}-k_{e}\right)}\left(e^{-k_{e}(t-t_{0})}-e^{-k_{a}(t-t_{0})}\right)$$
where *F* is the fraction of dose absorbed, *V_d_* is the volume of distribution, *k_a_* and *k_e_* are the absorption rate constant and elimination rate constant, respectively, *t* is the time of sampling, and *t*_0_ is the lag time. Then, if the *C_p_* found in adults is used and the pharmacokinetic parameters found in children (marked with′, such as *F*′) are added, the appropriate dose in children can be estimated.
(14)$$D_{child}=\frac{C_{p}V_{d}^{\prime }\left(k_{a}^{\prime }-k_{e}^{\prime }\right)}{F^{\prime }k_{a}^{\prime }\left(e^{-k_{e}^{\prime }t}-e^{-k_{a}^{\prime }t}\right)}$$

### 2.5. Physiologically Based Pharmacokinetic Calaculations

Using the information available in the literature, enzyme maturation and other parameters can be calculated that may help in calculating the clearance in children.
(15)$${Cl}_{children}={Cl}_{adults}{\left(\frac{Weight}{70\;kg}\right)}^{0.75}\times MFA$$

## 3. Results

### 3.1. Metabolism and CYP Enzyme Maturation

Calculated estimations or theoretical clearance rates of artemether and lumefantrine based on CYP enzyme maturation and liver development are presented in Table 3. The tissue water/fat ratio is not included in this calculation, but tissue distribution and tissue elimination have a significant influence on the clearance, which is not expressed in this table. Standard liver volume was calculated based on Walter et al. [60], and enzyme maturation was calculated based on Bonate et al. [54] and Johnson et al. [53].

### 3.2. Dosage Calculation

Table 4 shows the calculated dose based on different scaling methods for artemether and lumefantrine in healthy children from 2 months to 5 years.

### 3.3. Pharmacokinetic Method

According to the guidelines published by the WHO (WHO, 2015), the recommended dosage for artemether is 5–24 mg/kg but 29–144 mg/kg for lumefantrine. Since artemether and lumefantrine tablets are fixed solid dosage forms, dosing is per both the WHO and manufacturer guidelines. Infants and children weighing 5 kg to less than 15 kg receive 20 + 120 mg of artemether—lumefantrine. Children weighing 15 kg to less than 25 kg receive 40 + 240 mg of artemether—lumefantrine twice a day for three days. The second dose is given after 8 h and then every 12 h for a total number of three days. Using the lowest dose (20 and 120 mg/kg for artemether and lumefantrine, respectively) for an infant weighing 5 kg, the estimated pharmacokinetic profile for these two drugs may be seen in Figure 2 and compared with the standard dose for adults.

### 3.4. Physiological Information

Using physiological information to estimate the dose, including the maturation of metabolic enzymes and the pharmacokinetics of the drug, the estimated doses of artemether and lumefantrine to be used in children are suggested and compared with adult doses, as shown in Figure 3.

## 4. Discussion

### 4.1. Metabolism and CYP Enzyme Maturation

During development from fetus to adulthood, the liver undergoes development and metabolic enzymes take time to reach maturity. The size of the liver in infants and small children is relatively larger than that in adults. This may be reflected by variations in the activity of metabolizing enzymes with respect to liver size. Having a relatively larger organ will require relatively more blood flow, which will affect the disposition and can lead to either underexposure and treatment failure or overexposure with accompanying toxicity [39]. Cytochrome P450 (CYP) is a super-family of enzymes, categorized into various groups and subgroups based upon the similarity in the amino acid sequence. CYP3A, CYP2D6, and CYP2C9 together are responsible for almost 85% of the entire drug oxidation activities in people.

In embryonic life until late infancy, CYP3A7 is one of the important enzymes [61] when it comes to metabolizing drugs. Among the cytochrome P450 (CYP) enzymes, CYP3A predominates in the liver of adults and is responsible for the metabolism of almost 50–60% of the entire marketed drugs at present. The expression of CYP3A4 slowly increases during childhood until it surpasses that of adults and then slowly decreases to the level of adults at the end of puberty. The levels of this enzyme rise to 50% of the value found in grown-ups between 6 and 12 months of age, affecting the pharmacokinetics of these drugs [38,39,61,62]. Therefore, changes in drug clearance caused by these enzymes may require adjustment in doses and/or dosing intervals in small children.

When it comes to the metabolism of artemether, CYP3A4 and CYP2B6 are responsible for immediate hepatic metabolism [38,63,64]. Lumefantrine, however, is absorbed more slowly and is largely metabolized by CYP3A4 to its active metabolite [63,64].

### 4.2. Dosage Calculation

Children from 2 months to 5 years are the age group that is most at risk of being hit by severe or cerebral malaria. Malnourished children are a population group of special concern, since absorption from the GI tract may take much longer time than in normal children due to intestinal villous atrophy in the jejunal mucosa, as previously discussed. Due to poor absorption in malnourished children, these children may require higher dosages, such as those recommended by the WHO, to ensure that some of the drug is taken up by the GI tract. Therefore, lumefantrine has been shown to have bioavailability as low as 5% in such children.

In general, there is little information available for malnourished children when it comes to pharmacokinetic parameters, although it is known that they are highly affected by nutritional status. Thus, it is currently necessary to use the same dosage estimation as for non-malnourished children. In general, the calculated doses of artemether and lumefantrine based on body weight should be lower (i.e., 0.34 mg/kg for AR and 6 mg/kg for LF) than the doses recommended by the WHO for non-malnourished children. An exception is malnourished children, in whom researchers have reported significantly decreased absorption compared to that in healthy children.

### 4.3. Pharmacokinetic Method

When comparing a once-a-day dosage in children (Figure 3a,b), which is not recommended by the guidelines, to the recommended dosage (Figure 2e,f), the graphs follow the adult PK better. In malnourished children, protein manufacturing is significantly reduced, resulting in low serum albumin and low lipoprotein concentrations that will affect protein binding, especially for lumefantrine, which is 99.7% bound to circulating proteins. However, because of the nature of α1-glycoprotein, levels of this protein fluctuate based on health conditions, e.g., levels may increase in the gut due to inflammation resulting from the pathophysiological condition in the gut mucosa of malnourished children.

There are multiple other factors that influence the pharmacokinetics of artemether and lumefantrine in malnourished children, such as the effect of food, that have dramatic effects on the absorption of both drugs. The reason is that food increases the secretion of bile salts that will improve the absorption of these lipophilic drugs. The bioavailability of lumefantrine in fasted or malnourished children may be as low as 4.7% [65]. Other influencing factors in malnourished children are the villous atrophy as well as the reduced bile salt concentration secreted into the duodenum. These factors will be important in understanding what happens when infected with malaria and how the infection will change the health status and alter the pharmacokinetics of both drugs.

It has been described that malaria may increase the half-life of lumefantrine as in healthy individuals the terminal half-life is 2 to 3 days while it is 4 to 6 days in falciparum malaria patients. Therefore, this may affect the steady-state plasma concentration. These factors remain to be studied. Various methods are used to scale the dosage from adults to children. Accurate and effective dosing of patients depend on an accurate estimation of the PK and PD of the drug. In turn, the PK and PD of a drug are dependent on the patient’s physiological characteristics, which are unique for each patient as well as patient population.

Here, the model used was based on a one-compartment model, whereas real data showed that both drugs, artemether and lumefantrine, followed a two-compartment model, as seen in Figure 2a,b. However, pharmacokinetic values for artemether and lumefantrine in children are scarce, and some of the values have significant effects on the dose selected, bioavailability, volume of distribution, and elimination half-life. In addition, there is little information available for malnourished children when it comes to pharmacokinetic parameters, although it is known that these parameters are highly affected by nutritional status. This makes it difficult to estimate the accurate dose to be used in the development of a new formulation for artemether and lumefantrine for children.

### 4.4. Physiological Information

Physiological changes happen during the growth of a child. Body composition affects various parameters, such as the volume of distribution (*V_d_*), the maturation of the liver and its enzymes, and how this affects the clearance and half-life of the drug as well as the plasma protein levels for different proteins that bind to the compound of interest. Lerman et al. [66] observed a 50% reduction in the concentration of AAG in the serum of preterm neonates compared to that of full-term neonates, while it was 300% higher in infants than in full-term neonates. However, from infants to adolescents, the AAG concentration only rose by 25%.

This indicates that the free fraction of drugs that binds to AAG may be highest in neonates and decline with an increase in age. Therefore, a reduction in plasma protein binding is not only because of a decrease in the overall amount (quantity) of plasma proteins, but there may also be a reduction in the binding affinity (quality) that affects the overall binding ability [15], resulting in altered pharmacokinetics and pharmacodynamics of drugs compared to adults. Having a high free drug concentration may, however, have its drawbacks, such as altered volume of distribution, clearance, and half-life, since free drug is cleared more easily than drug bound to plasma proteins and red blood cells. In severe cases of malaria, such as cerebral malaria, drug distribution to the brain is required.

Lipid-soluble drugs have more easy access across the blood-brain barrier (BBB), and this access is even easier in very young children compared to older ones and adults. This is because the BBB has not matured enough to prevent drugs from penetrating across and into the brain [67]. The manufacturing of these plasma proteins is highly affected by the nutritional status of the individual, so children that are malnourished may have a decreased concentration of these proteins [68]. Human serum albumin (HSA) is the most abundant protein in plasma and binds to most drugs, followed by α_1_-acid glycoprotein [66,68]. The normal plasma concentrations of both proteins are lower in neonates and infants than in older children [66], so the effect of malnutrition is more dramatic in young children. Therefore, an increase in the free fraction of the drug may be expected in children with accompanying consequences.

This information can be used to estimate the dose. However, there are not many pharmacokinetic parameters available for children, making a real extrapolation difficult. For lumefantrine, the published *V_d_* is very high for children and the bioavailability is very low, especially for children that take the medicine in a fasted state. These factors have a dramatic effect on the pharmacokinetics if they are true. If the *V_d_* is as high as described here, the dose needs to be significantly increased to obtain a similar plasma concentration as in adults. Some authors have claimed that patient age and gender have no effect on any of the main pharmacokinetic parameters, such as absorption rate constant (k_abs_) and Cl [69].

According to Ashley et al. (2007), the absorption of lumefantrine may increase 6-fold when taken with soy milk. In the case of artemisinin combination therapies (ACTs), poor absorption of lipids and fats due to villous atrophy can affect the absorption of lipid-soluble ACTs [70]. Decreased exposure to lumefantrine has been reported in children with severe acute malnutrition [47,71,72]. In addition, the metabolism of antimalarial drugs can be decreased due to low plasma albumin and other plasma proteins concentrations, as previously explained [72]. Hypoalbuminemia is more acute in kwashiorkor than in marasmus [47,71,72]. Hepatic metabolism of antimalarial drugs can be dysregulated and decreased or increased depending on whether the malnutrition is predominantly a protein deficiency or global malnutrition [73]. Therefore, a deeper understanding of different types of malnutrition and their effects on pharmacokinetics is important so that pediatric doses can be adjusted accordingly [47,71,72].

The monitoring of hepatic-metabolized antimalarials is, hence, crucial in children with malnutrition. Interstitial water retention is another critical pathophysiological change in SAM that also affects the pharmacokinetics of antimalarial drugs [47]. Water retention increases the volume of distribution of drugs and, in turn, can cause suboptimal drug exposure of a standard dose [47,72]. Children with undernutrition risk dosing inaccuracies such as underdosing for weight when the dosing is only based on age, or overdosing for age when the dosing is based on weight alone [40]. Dose optimization for ACTs, for the recommended treatment for uncomplicated *P. falciparum* malaria in most malaria-endemic countries, is crucial for optimum dosing efficacy, safety, and avoiding parasite resistance [74].

A recent meta-analysis showed a higher risk of treatment failure in malnourished children than in well-nourished children in Africa [75]. Furthermore, during treatment with AR-LF for uncomplicated malaria, younger children under 3 years, including underweight-for-age children, had a 23% lower concentration of the drug adjusted for mg/kg at day 7 in comparison with well-nourished children of the same age [48]. To optimize the dose for malnourished children, pharmacokinetic and pharmacodynamic data are essential. In tackling the question of the effect of SAM on the pharmacokinetics and pharmacodynamics of artemether-lumefantrine for the treatment of uncomplicated malaria in children, a study in 2019 showed essential findings [76]. The study found that all malnutrition was associated with reduced absorption of lumefantrine [76]. Specifically, the mid-upper-arm circumference (MUAC) was the most significant covariate, indicated by a 25.4% decreased absorption per 1 cm reduction in MUAC [76]. In addition, the study depicted that the lower exposure of lumefantrine resulted in an increased risk of acquiring a new *Plasmodium falciparum* infection during the follow-up period [76]. Such results emphasize the essential need for dose optimization for ACTs in children with SAM to control for the poor absorption and achieve the intended efficacy in these children at risk of the adverse effects of malaria and malaria re-infection.

## 5. Conclusions

Individual variations in patients in a defined patient population affect the PK and PD of drugs. Children under the age of 5 years are not small adults, and their physiology, liver enzyme maturation, and serum proteins have not fully developed and may, therefore, significantly influence the pharmacokinetics of drugs and how they behave in the body. Obese children have a greater quantity of body fat, which may lead to an increased volume of distribution of lipid-soluble drugs [77]. Therefore, when designing a dose regimen, it is important to know and include the distinctions in physiology at various phases of development that influence the pharmacokinetics of various drugs when estimating the dose in young children. The possibility of estimating the accurate dose in children depends on the availability of more pharmacokinetic data derived from clinical studies.

Malnutrition, lack of clean water, and infections (e.g., helminthics) may affect the pharmacokinetics of drugs in children when they need to receive acute treatment, such as for malaria. It is, therefore, important to understand how these factors affect how the drug is handled by children living in parts of the world where severe and cerebral malaria are endemic. Dose adjustments in children that are categorized as having normal weight, compared with obese or malnourished children, need to be based on the pharmacokinetic changes that happen in these situations. Therefore, accurate and effective dosing of patients depend on an accurate estimation of the pharmacokinetics (PK) and pharmacodynamics (PD) of the drug.

Due to the lack of real pharmacokinetic data for children in the literature, a clinical study needs to be conducted to verify whether the suggested doses can be clinically efficacious.

## Figures and Tables

**Figure 1 pharmaceutics-15-01076-f001:**
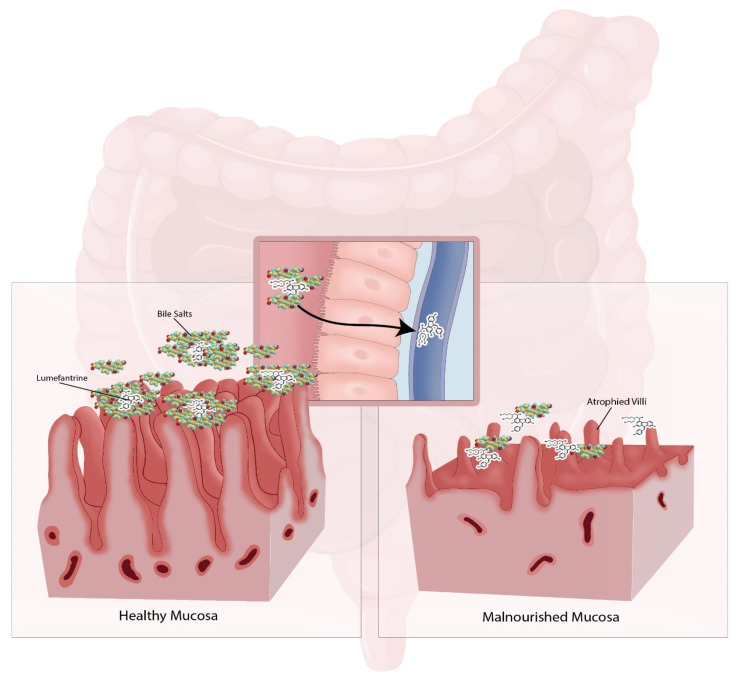
The intestinal mucosae in healthy and malnourished subjects, showing the atrophy of villi in the jejunum as well as the reduced amount of bile, which is required to augment the absorption of compounds such as lumefantrine.

**Figure 2 pharmaceutics-15-01076-f002:**
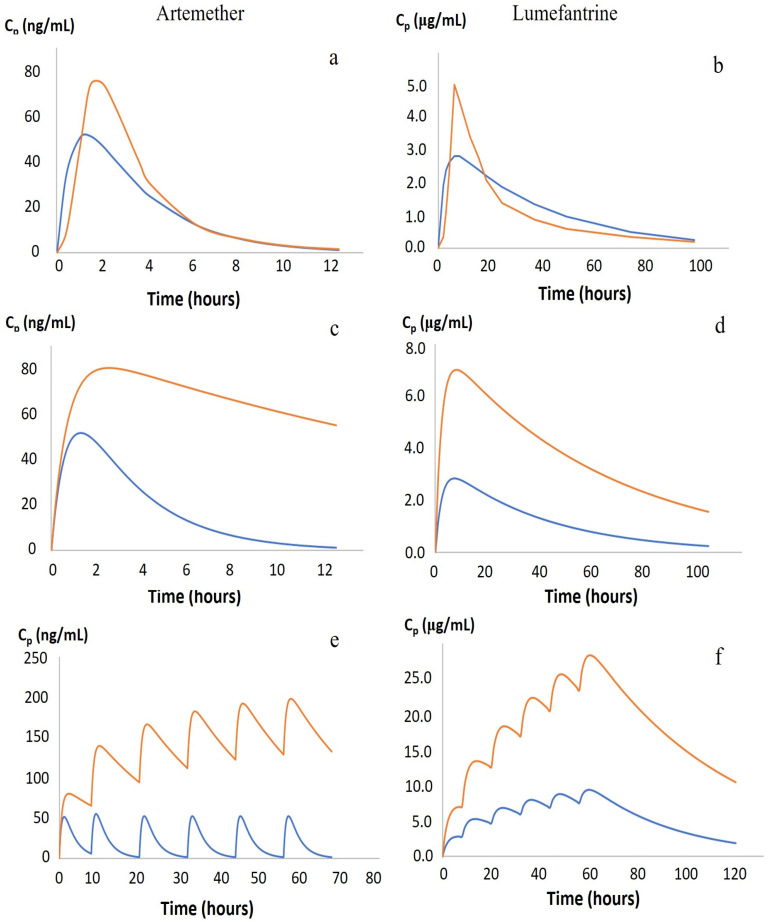
Plasma concentration profiles of artemether and lumefantrine. (**a**,**b**) The calculated pharmacokinetics in adults (blue) in comparison with real adult data based on the pharmacokinetics of Coartem^®^, from Novartis (Basel, Switzerland) (orange). (**c**–**f**) The pharmacokinetics in adults (blue) and a 5 kg child (orange), using pharmacokinetic values from the literature and dosed according to the guidelines from the World Health Organization. (**c**,**d**) A single dose administration of artemether (**c**), and lumefantrine (**d**). (**e**,**f**) The appearance of full treatment when doses are administered at time 0, 8 h, then every 12 h for 3 days for artemether (**e**), and lumefantrine (**f**).

**Figure 3 pharmaceutics-15-01076-f003:**
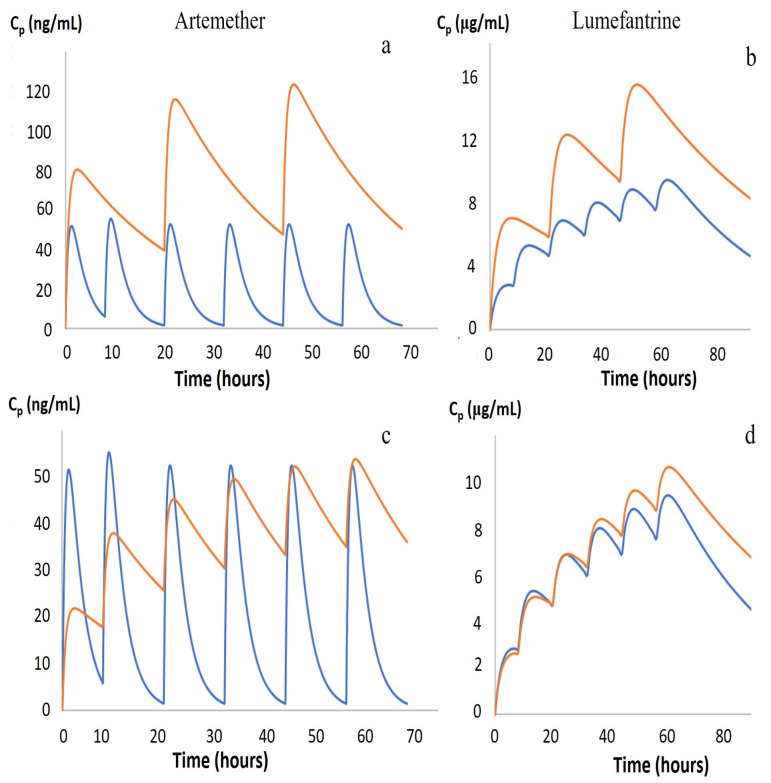
Plasma concentration time profiles of artemether and lumefantrine in adults and children. In (**a**,**b**), the dose-regimen has been changed for children in such a way that children take their medicine once a day, not at times 0, 8 h, then every 12 h for 3 days, where the pharmacokinetics for artemether (**a**) and lumefantrine (**b**) are shown. In (**c**,**d**), the dose of artemether has been adjusted for children to 0.34 mg/kg instead of 5–24 mg/kg (**c**); the dose for lumefantrine has been adjusted to 6 mg/kg instead of 29–244 mg/kg (**d**).

**Table 1 pharmaceutics-15-01076-t001:** The concentrations of selected plasma components in different population groups.

Plasma Component	Adults	Newborn	Infant	Child	Malnourished Children	Drug Binding
Plasma albumin (g/L)	35–50	29–54	44–54	40–58	↓↓ (<24.5)	Artemether
α_1_-glycoprotein (g/L)	1.0–4.0	1.0–3.0	2.0–4.0	1.0–4.0	↑↑	Artemether
Cholesterol (mmol/L)	<5.2	1.9	2.3–3.4	3.4–4.4	↓↓	
LDL (mmol/L)	1.55–4.14	0.78	2.07	2.5	↓↓	Lumefantrine
HDL (mmol/L)	1.52–4.05	1.6	2.6	3.4	↓↓	Lumefantrine
VLDL (mmol/L)	0.1–1.7	0.28	0.62	0.45	↑↑	Lumefantrine
Triglycerides (mmol/L)	0.44–2.09	0.46	0.05–0.46	0.88–1.56	↑↑	Lumefantrine
RBCs (×10^12^/L)	4.0–6.0	4.8–7.2		3.8–5.5	↑↑	Artemether/lumefantrine

↑↑ = increased plasma component, ↓↓ = decreased plasma component.

**Table 2 pharmaceutics-15-01076-t002:** The pharmacokinetic parameters for artemether and lumefantrine, based on values obtained from the World Health Organization (Guidelines for treatment of malaria, 3rd edition) and literature.

	Artemether			Lumefantrine			
Parameter	Adults	Children	Malnourished Children	Adults	Children	Malnourished Children	Comments
Dose (mg/kg)	1.14	5		6.86	29		
C_max_ (ng/mL)	100	119		7.91 **		15% lower	Malaria (and food) increases C_max_ for AR 2-3 fold
t_max_ (h)	2.0			6			
t_1/2_ (h)	1.9	16		95	123	12% shorter	Malaria makes half life significantly longer (LF)
AUC (ng·h/mL)	320	392		207		31% lower	Malaria increases AUC significantly (AR)
Vd (L/kg)	6.05	0.9		3.8/0.71	1.3		
Cl (L/h per kg)	0.91						3 fold lower Cl in <3 months vsv >3 months (AR)
k_a_ (/h)	1.52	1.2		0.44	0.46		Malaria lowers k_a_ significantly (LF)
F (%)	43			65			Lower in fasted subjects, or age <5 year (LF)
fu (%)	4.6			0.3			
**Drug Interactions**							
Effect of ketoconazole	↑			↑			
Effect of mefloquine	NE			↓			
Effect of quinine	↓			NE			
Effect of lopinavir/ritonavir	↓			↑			

** Food may increase the absorption of lumefantrine up to 16—fold due to increase in bile salts. Units for LF are (µg/mL). NE = No effect, AUC = Area Under the Curve, V_d_ = volume of distribution, t_max_ = Time to maximum plasma concentration, Cl = Clearance, ↑ = increased plasma concentration or exposure, ↓ = decreased plasma concentration or exposure.

**Table 3 pharmaceutics-15-01076-t003:** Theoretical calculation of clearance for lumefantrine and artemether based on liver enzyme maturation in children, using the elimination rate constant in adults as a reference value.

	Standard Liver	Liver Volume	Enzyme Maturation		Lumefantrine	Artemether
Age	Volume (mL)	As % weight	CYP3A4	CYP2B6	Clearance (L/h/kg)	Clearance (L/h/kg)
2 mo	162	3.06	0.18	0.12	0.42	5
6 mo	198	2.60	0.48	0.30	0.99	15
1 y	223	2.40	0.78	0.48	1.53	27
2 y	258	2.19	0.96	0.65	1.77	37
3 y	287	2.03	1.09	0.74	1.89	45
5 y	334	1.82	1.09	0.85	1.75	51
Adult	1.648	2.35	1.00	1.00	1.94	154

**Table 4 pharmaceutics-15-01076-t004:** Calculated dosages for children, at different ages, using various published methods to extrapolate from adult dose to pediatric dose, compared to WHO recommendations for the same age.

**Artemether (mg)**							
**Age**	**Weight (kg)**	**Height (cm)**	**WHO**	**Young’s Rule**	**Clark’s Rule**	**Area Rule**	**Allometric Scaling**
2 mo	5.3	57.3	20	1.1	6.2	13	17
6 mo	7.6	66.6	20	3.2	8.9	17	23
1 y	9.3	75	20	6.2	10.9	20	26
2 y	11.8	87.3	20	11.4	13.9	25	32
3 y	14.1	95.5	20	16.0	16.6	28	36
5 y	18.3	109.8	40	23.5	21.5	35	44
**Lumefantrine (mg)**							
**Age**	**Weight (kg)**	**Height (cm)**	**WHO**	**Young’s Rule**	**Clark’s Rule**	**Area Rule**	**Allometric Scaling**
2 mo	5.3	57.3	120	6.6	37.4	81	104
6 mo	7.6	66.6	120	19.2	53.6	104	136
1 y	9.3	75	120	36.9	65.6	122	158
2 y	11.8	87.3	120	68.6	83.3	148	189
3 y	14.1	95.5	120	96.0	99.5	170	216
5 y	18.3	109.8	240	141.2	129.2	207	263

## Data Availability

The data generated during this study are available from the corresponding author upon request.

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
