# Peer review of "Estimation of Pediatric Dosage of Antimalarial Drugs, Using Pharmacokinetic and Physiological Approach"

_pharmaceutics, 2023, doi:10.3390/pharmaceutics15041076_

Round 1

Reviewer 1 Report

I would like to warmly congratulate the Authors for their important contribution to pediatric malaria therapy optimization. I have only a very minor comment, about the word allometry (line 327). It should be classically used just to describe and translate PK/PD results from a preclinical to a clinical context, and not from human to human as in your dedicated chapter. Maybe a synonymous could better fit this topics 

Author Response

This comment is highly appreciated. See attached document.

Reviewer 2 Report

Comments to Authors

    The manuscript discussed an important issue which is doses estimation in children using the physiological information from children and some pharmacokinetic data from adults. The manuscript aimed to estimate the optimal doses that can mimic adult exposure. Certain modifications are required to be performed through the following comments; 

1-      Paragraphs SHOULD NOT exceed 8 to 10 lines throughout the manuscript in order not to lose readers attention and interest.

21-   Please summarize the paragraphs at page lines 62 - 89 in the introduction as it is too long.

I 2- will be much better if a study flow chart be added showing the different resources  of data recruitment including the websites, researches …..etc , as well as, the different procedures followed in the study. 

43-The authors SHOULD write about the pathophysiology, sign and symptoms, epidemiology of malaria disease in children. Moreover, the mechanism of actions as well as reasons for selecting these two main medications, namely, Lumefantrine and Artemether as candidates for this research.

5 4- Page 5 line 198 under the subtitle nutritional status the authors mentioned several times undernutrition and malnutrition without mentioning the types of food or liquids required to overcome the incidence of undernutrition. The authors had to add a table showing the ideal diets recommended for children at their several stages of growth. The following paragraph is recommended to be added.

6 5- Healthy diet including fruits and vegetables, can supply the body with beneficial nutrients and antioxidants (1) including coenzyme Q10 and alpha-tocopherol have proved to have a protective effects of antioxidants (2,3).

References

(1) Sara AR, Mohamed Raslan, Eslam M Shehata and Nagwa A Sabri. Impact of Applied Protective Measures of COVID-19 on Public Health. Acta Scientific Pharmaceutical Sciences 5.7 (2021):63-72.

(2) https://doi.org/10.1016/j.neuro.2018.07.006.

(3) https://doi.org/10.1007/s12640-018-9971-6

7 6- Information concerning the effect of comorbidities on the pharmacokinetics of the candidate drugs as well as other antimalarial medication was missed. The authors SHOULD add this important information in the manuscript

8 7-The effect of any co-administered medications in children either in chronic diseases or acute cases on the pharmacokinetics of Lumefantrine and Artemether was missed as well. The authors SHOULD write an updated paragraph in this issue.

9 8- The need of supplements for child growth including vitamin D for example, are to be mentioned in the introduction under nutritional status section as low blood 25(OH)D levels have been reported in patients affected by infectious diseases caused by parasites, including malaria and other diseases as well. The following paragraph is recommended to be added; “Vitamin D is one of essential vitamins which was not only deficient in autistic children but also contribute to pathogenesis of the disease in these children (3)”.

Reference

DOI: 10.13040/IJPSR.0975-8232.7(3).1043-49.

9-Drug interaction as a very important issue involved in changes of pharmacokinetics of Lumefantrine and Artemether SHOULD be added in the manuscript.

1 10-Figures containing the different pathways and mechanisms by which the pharmacokinetics of Lumefantrine and Artemether are influenced by in children SHOULD be added.

       Comments to Authors

 The manuscript discusses an important issue which is doses estimation in children using the physiological information from children and some pharmacokinetic data from adults. The manuscript aimed to estimate the optimal doses that can mimic adult exposure. Certain modifications are required to be performed through the following comments; 

1-Paragraphs SHOULD NOT exceed 8 to 10 lines throughout the manuscript in order not to lose readers attention and interest.

2- Please summarize the paragraphs at page lines 62 - 89 in the introduction as it is too long. 

3-It will be much better if a study flow chart be added showing the different resources  of data recruitment including the websites, researches …..etc , as well as, the different procedures followed in the study.

4- The authors SHOULD write about the pathophysiology, mechanism of actions as well as reasons for selecting these two main medications, namely, Lumefantrine and Artemether as candidates for this research.

5-  Page 5 line 198 under the subtitle nutritional status the authors mentioned several times undernutrition and malnutrition without mentioning the types of food or liquids required to overcome the incidence of undernutrition. The authors had to add a table showing the ideal diets recommended for children at their several stages of growth. The following paragraph is recommended to be added.

6-  Healthy diet including fruits and vegetables, can supply the body with beneficial nutrients and antioxidants (1) including coenzyme Q10 and alpha-tocopherol have proved to have a protective effects of antioxidants (2,3).

References

(1) Sara AR, Mohamed Raslan, Eslam M Shehata and Nagwa A Sabri. Impact of Applied Protective Measures of COVID-19 on Public Health. Acta Scientific Pharmaceutical Sciences 5.7 (2021):63-72.

(2) https://doi.org/10.1016/j.neuro.2018.07.006.

(3) https://doi.org/10.1007/s12640-018-9971-6

 7-Information concerning the effect of comorbidities on the pharmacokinetics of the candidate drugs as well as other antimalarial medication was missed. The authors SHOULD add this important information in the manuscript.

8-The effect of any co-administered medications in children either in chronic diseases or acute cases on the pharmacokinetics of Lumefantrine and Artemether was missed as well. The authors SHOULD write an updated paragraph in this issue.

9-      The need of supplements for child growth including vitamin D for example, are to be mentioned in the introduction under nutritional status section as low blood 25(OH)D levels have been reported in patients affected by infectious diseases caused by parasites, including malaria and other diseases as well. The following paragraph is recommended to be added; “Vitamin D is one of essential vitamins which was not only deficient in autistic children but also contribute to pathogenesis of the disease in these children (3)”.

Reference;

DOI: 10.13040/IJPSR.0975-8232.7(3).1043-49.

10-  Drug interaction as a very important issue involved in changes of pharmacokinetics of Lumefantrine and Artemether SHOULD be added in the manuscript.

11-  Figures containing the different pathways and mechanisms by which the pharmacokinetics of Lumefantrine and Artemether are influenced by in children SHOULD be added.

12-  A summarized collective table showing the different rules and methods that might be used in dose calculation showing the pros and cons of each with references SHOULD be added in order to simplify better the presented data to the readers.

13-  Methodologies to be used for patients’ follow-up (suggested therapeutic drug monitoring) after giving them the suggested doses was not given, suggested or cited in the manuscript. This SHOULD be added in order to explain the different procedures to be followed for ensuring that the suggested doses were the right ones and best choice evidenced by patients’ response.

1 14-A summarized collective table showing the different rules and methods that might be used in dose calculation showing the pros and cons of each with references SHOULD be added in order to simplify better the presented data to the readers.

115- Methodologies to be used for patients’ follow-up (suggested therapeutic drug monitoring) after giving them the suggested doses was not given, suggested or cited in the manuscript. This SHOULD be added in order to explain the different procedures to be followed for ensuring that the suggested doses were the right ones and best choice evidenced by patients’ response.

1 16-Conclusion has to be written in more informative way showing the impact of the authors proposals on the special population healthcare as well as the significance of its application from the scientific point of view.

Author Response

Your comments are highly appreciated. Please find responses to each individual comment in attached file.

Reviewer 3 Report

This manuscript estimated the optimal dosage for the two antimalarial drugs, artemether and lumefantrine, in young children under 5 years of age using physiological pharmacokinetic approach. Generally, the results could be valuable for other researchers and physicians in this field. Some comments were suggested as follows.

1. The abstract should state briefly the purpose of the research, the principal results and major conclusions. The current Abstract should be improved.

2. Keywords: Too many keywords were listed and obviously some of them were not eligible keywords. 

3. The presented introduction is too interminable to understand the significance and objectives of this work. I strongly recommend the authors refine this section. In addition, the relevant references to support the listed concentration of selected plasma components in different population groups should be included in Table 1.

4. 2.2. Metabolism and CYP enzyme maturation: It is unclear how to get the equations (2), (3), and  (4)? Plus, please recheck this statement: “one is able to calculate the overall degree of enzyme maturation (MFA) in children below the age of 25 years by adding the outcome into following equation [2]” following equation [2]? In addition, please clarify how to determine liver weight.

5. Figure 2 and 3: Figure 2 and 3 should be redrawn with the unit added to the x-axis and y-axis.

6. It should be noted that all the results reported in this work were estimated using the physiological information from children and pharmacokinetic data from adults. The authors should at least add real clinical PK data obtained from pediatric clinical practice to compare and verify the reliability and applicability of the reported approach in this work.

Author Response

(The authors gave the same response as above.)

Round 2

Reviewer 3 Report

The authors have addressed my concerns in this revision. I have no more comments.